# Influence of Health Related Fitness on the Morphofunctional Condition of Second Mature Aged Women

**DOI:** 10.3390/ijerph17228465

**Published:** 2020-11-16

**Authors:** Olha Podrihalo, Svetlana Savina, Leonid Podrigalo, Sergii Iermakov, Władysław Jagiełło, Łukasz Rydzik, Wiesław Błach

**Affiliations:** 1Department of Biological Science, Kharkiv State Academy of Physical Culture, 61022 Kharkiv, Ukraine; rovnayaolga77@ukr.net; 2Department of Dance Sports, Fitness and Gymnastics, Kharkiv State Academy of Physical, 61022 Kharkiv, Ukraine; info.fitness.ukr@gmail.com; 3Department of Medical Science, Kharkiv State Academy of Physical Culture, 61022 Kharkiv, Ukraine; l.podrigalo@mail.ru; 4Department of Sport, Gdansk University of Physical Education and Sports, 80-336 Gdansk, Poland; sportart@gmail.com (S.I.); wjagiello1@wp.pl (W.J.); 5Institute of Sport Sciences, University of Physical Education in Krakow, 31-541 Kraków, Poland; 6Department of Sport, University School of Physical Education, 51-612 Wroclaw, Poland; wieslaw.judo@wp.pl

**Keywords:** complex fitness program, stretching, joints’ mobility, morphofunctional indicators

## Abstract

To analyze the influence of health-related fitness on the condition of second mature aged women. Participants: 65 women divided into two groups. Group 1–40 women, (43.33 ± 0.93) years old and group 2–25 women (44.40 ± 0.93) years old. The participants trained for 8 months, three times a week for 1 h. Group 1 trained dance aerobics (Monday), strength fitness (Wednesday) and stretching (Friday). Group 2 trained only stretching. The body length and mass, handgrip strength test, vital capacity, blood pressure, heart rate, Stange and Genchi tests, and motion amplitude in joints were evaluated before and after the program. The significance of the differences between the groups was evaluated by Student’s criterion (t) and Rosenbaum (Q). The different intensity of the health-related effect was confirmed at the end of the program. Physiometric indicators significantly increased in group 1. The complex physical activity led to a decrease in heart rate. The results of the Stange and Genchi tests significantly increased. Goniometric indicators of group 2 increased. The comparative analysis of the participants indicators confirms the generalized and higher health-related effect of the complex fitness program. The effect of such a program showed an increase of the adaptive potential, a significant increase in the functional capabilities of women, and the optimization of the studied indicators. With the same time expenditure for health-related fitness, the complex program has a more multifaceted effect in comparison with stretching.

## 1. Introduction

The optimal level of physical activity is a leading factor in maintaining health. Checking motor activity allows for controlling your health and ensuring its adequate monitoring. Numerous population studies confirm the positive value of recreational activities for improving health [1,2] and the prevention of chronic non-communicable disease [3,4,5].

Some of the studies emphasize the importance of using social networks and mobile applications in organizing the students’ functional condition monitoring [6]. The most informative evaluation criteria are control of body mass and the tolerance of dosed physical loads.

A high level of physical fitness can positively influence both health and cognitive functions. Therefore, monitoring its level at a young age allows for prediction, control changes, and control of the health level and life quality of the adult population. The systematic review aims to identify field tests used by youth involved in sports; a search for the necessary control tools is presented [7].

The study by Ponomareva is dedicated to finding new means and methods of physical education to improve students’ physical fitness and health [8].

Another recent study emphasizes the importance of physical education for students. The purpose of such fitness classes is the formation of psychophysiological readiness for future professional activities [9].

Recommended forms and means of physical training transformed depending on age. So, moderate-intensity endurance training is recommended for elderly people to improve their metabolic health [10]. These exercises cause a favorable adaptation of skeletal muscles due to an increase in mitochondria. This can help maintain muscle oxidative ability and slow down the aging process.

Some researchers emphasize the importance of using exercise programs for older women. Such programs have reduced anxiety and depression and developed flexibility, tolerance to aerobic exercise, general endurance, coordination, the balance of dexterity, and dynamic balance [11].

Another study showed the importance of analyzing physical development and somatotype features in girls and females involved in dance sports and gymnastics. It confirmed the informative value of the bioimpedance method for predicting the athlete’s condition [12].

The correlation between body composition, indicators of the autonomic system, adaptive capabilities of the cardiorespiratory system, level of motor activity, and impaired neuropsychic status was presented in another research. The designed regression equations are proposed to predict the condition of 26–43 year-old women. The equations allow for evaluating the risks of functional disorders in overweight women [13].

Similar results were obtained by several researchers who confirmed the existence of a connection between general and special physical fitness in women involved in judo, and age-somatic criteria [14].

In another research its authors used a complex of anthropometric, physiological indicators and indices, and the results of stress tests to predict physical performance. It is proposed to use the designed regression equations in the tolerance analysis of physical loads [15].

Similar results are reported by Vasquez–Gomez et al. The authors designed a prediction of the students’ physical performance based on anthropometric indices, indicators of the cardiovascular system, and the results of a six-minute walk test [16].

Thus, the available literary sources confirm the relevance of studying the effectiveness of health-related fitness programs for the second mature aged women. The specificity of this age group is that the level of work capacity is still quite high, but physiological changes are also intensified, leading to accelerated aging and a decrease in the health level. The purpose of this study is a comparative analysis of the influence of health-related fitness carried out according to different programs on the morphofunctional condition of the second mature aged women.

## 2. Materials and Methods 

### 2.1. Participants 

The study involved 65 women, divided into two groups. Group 1–40 women, average age (43.33 ± 0.93) years old and group 2–25 women, average age (44.40 ± 0.93) years old. The participants belonged to the group of mental and light physical labor workers. There were no significant differences in marital status or the number of children between the groups.

All participants engaged in health-related fitness for 8 months, three times a week for 1 h. Group 1 had a complex program that included dance aerobics (Monday), strength fitness (Wednesday) and stretching (Friday). The classes consisted of three parts: introductory (5–7 min), main (40–47 min), and final (8–13 min). The introductory part consisted of warm-up exercises for the main muscle groups. The main part of strength fitness included exercises with dumbbells weighing 1–2 kg or a body bar. In dance aerobics classes, dance exercises were performed to music. Stretching exercises were performed in the stretching class. The final part included breathing exercises to relieve emotional and motor arousal.

Participants in group 2 were engaged only in stretching.

### 2.2. Study Design 

The study design involved determining the morphological and functional indicators before and after the program. The measurements were carried out according to the requirements of the international unified methodology of anthropometric studies [13]. The length and body mass, handgrip strength test, vital capacity, blood pressure, and heart rate was measured. The body mass index was calculated as the ratio of mass (kg) to body length squared (m^2^). Pulse pressure was calculated as the difference between systolic and diastolic blood pressure. The timed inspiratory capacity (Stange test, s) and timed expiratory capacity (Genchi test, s) were evaluated. The motion amplitude in joints was measured using a mechanical goniometer Kawe (Kirchner & Wilhelm GmbH+Co.KG Eberhardstr. 56 71679 Asperg Germany). The following amplitudes (degrees) were also measured: shoulder flexion (right); shoulder flexion (left); shoulder extension (right); shoulder extension (left); shoulder abduction (right); shoulder abduction (left); hip flexion (straight leg) (right); hip flexion (straight leg) (left); hip extension (right); hip extension (left); torso flexion. 

### 2.3. Statistical Analysis 

Statistical analysis of the obtained data was carried out by licensed MS Excel (licensed package of applied programs "Microsoft Office Professional" BGW7766 269-01611). Indicators of descriptive statistics were determined: arithmetic mean, standard deviation, and mean error. The significance of differences in the groups was evaluated by the Student’s parametric criterion (t) and non-parametric criteria of signs (z) and Rosenbaum (Q). The dynamics of indicators in each group was evaluated according to the criteria t and z, the groups were compared according to the criteria t and Q.

### 2.4. Ethics Statement

This study was approved by the Bioethics Committee (protocol of the Commission on Bioethics of the Kharkov State Academy of Physical Culture 23.05. 2018, No. 5) and conducted according to the Declaration of Helsinki. All participants gave their written consent to the research and were informed about the purpose and test procedures and about the possibility of withdrawal of consent at any time for any reason.

## 3. Results

The obtained data are shown in Table 1. The differences in the morphofunctional indicators of the participants depending on the training program were determined. The complex organization of classes in group 1 led to the optimization of many indicators. A significant increase in handgrip strength test in the dynamics of classes was shown. Differences are confirmed for the right hand (t = 3.17, z = 1), and the left hand (t = 2.63, z = 4). Similar changes were determined by the vital capacity (VC, z = 7). The following values decreased: systolic blood pressure (z = 5), diastolic blood pressure (z = 4), and pulse blood pressure (z = 7), resting heart rate (t = 2.98, z = 1). The time for Stange test (t = 2.70, z = 1) and Genchi test (t = 4.19, z = 0) significantly increased.

The dynamics of changes in group 2 increased significantly less. Student’s parametric criterion did not show significant differences in any indicators; i.e., did not confirm their dynamics during the 8 months of training. The nonparametric criterion of signs was more informative in this case. Changes were observed only in the case of some of the participants. The results showed a significant increase in handgrip strength test (right) (z = 2, n = 16) and handgrip strength test (left) (z = 3, n = 18). A decrease in body mass index (z = 7, n = 25), heart rate (resting heart rate (z = 4, n = 17) was observed. It was confirmed that the improvement of Stange test (z = 3, n = 21) and Genchi test (z = 3, n = 18) took place.

The intergroup comparative analysis at the end of the program confirmed the differing intensity of the health related effect. Physiometric indicators were significantly higher in group 1. This is confirmed for handgrip strength test (right) (t = 4.50; Q = 18) and handgrip strength test (left) (t = 6.15; Q = 17), VC (t = 2.40; Q = 12). Complex physical activity led to a decrease in resting heart rate (t = 3.19) and normalization of pulse blood pressure (Q = 12). The results of Stange test (t = 3.86; Q = 11) and Genchi test (t = 2.97; Q = 8) significantly increased.

Both programs included targeted activities to develop flexibility. This led to an improvement in this quality by increasing the amplitude of motions, increasing mobility in the joints. The significant increase in indicators by parametric and nonparametric criteria were confirmed in both groups. The following significant increases were confirmed In group 1: in the amplitude of the shoulder flexion (right) (t = 4.24, z = 0) and the amplitude of the shoulder flexion (left) (t = 4.87, z = 0), shoulder extension (right) (t = 4.23, z = 0) and shoulder extension (left) (t = 3.87, z = 0), shoulder abduction (right) (t = 3.44, z = 0) and shoulder abduction (left), (t = 3.72, z = 0), hip flexion (straight leg) (right) (t = 3.99, z = 0) hip flexion (straight leg) (left) (t = 4.09, z = 0), hip extension (right) (t = 4.53, z = 0) and hip extension (left) (t = 4.81, z = 0), as well as torso flexion (t = 3.12, z = 0).

Similar changes were observed in group 2. The significant increase in the following parameters were confirmed: the amplitude of shoulder flexion (right) (t = 3.49, z = 0) and shoulder flexion (left) (t = 3.91, z = 0), shoulder extension (right) (t = 3.15, z = 0) and shoulder extension (left) (t = 3.08, z = 0), shoulder abduction (right) (t = 3.79, z = 0) and shoulder abduction (left) (t = 3.68, z = 0), hip flexion (straight leg) (right) (t = 4.48, z = 0) and hip flexion (straight leg) (left) (t = 4.91, z = 0), hip extension (right) (t = 3.30, z = 0) and hip extension (left) (t = 3.44, z = 0), as well as torso flexion (t = 3.02, z = 0).

The intergroup comparison of results at the end of the program also revealed certain differences. The amplitude of shoulder extension (right) was higher in group 1 (Q = 7). The goniometric indicators of group 2 were higher in the shoulder abduction (right) (t = 4.50; Q = 17) and the shoulder abduction (left) (t = 4.91; Q = 23), hip flexion (straight leg) (right) (t = 2.03, Q = 7) and hip flexion (straight leg) (left) (t = 2.56, Q = 7), hip extension (left) (t = 2.04), and torso flexion (t = 2.60).

## 4. Discussion

Analysis of the influence of physical stress on the body in sports and health-related physical culture pursues various aims. In sports, it aims to identify the specific influence of a particular activity on the body [17]. The main issue is to increase the adaptive potential, and to maintain and increase the health level of students in health-related physical culture. The selection of criteria for assessing the fitness classes’ effectiveness was carried out considering the focus of the program. The analyzed dynamics of morphological and functional indicators confirm the increase in basic physical qualities (strength, endurance, flexibility) and participants’ functional capabilities. The use of the goniometry technique is stipulated by its adequacy and informativeness for assessing flexibility. An increase in the amplitude of motions and mobility in the main joints is an important indicator of the normal condition of the musculoskeletal system. The possibility of applying this technique to monitor the functional condition of athletes engaged in different types of martial arts was studied by Podrigalo et al. [18]. 

The results of studies by other researchers confirm the adequacy of the use of physiological indicators to monitor the condition of people during physical activities. The study confirmed the informativeness of heart rate, respiratory rate, reliability, and validity [19].

In an additional review its authors emphasize the fact that only a complex of various tests allows a qualitative analysis of physical fitness. The complex is recommended to include tests to assess muscle strength and endurance, speed, agility, flexibility, and aerobic productivity [7].

In a research where a range of tests was used, the analysts evaluated the functional condition of the respiratory system of female students. The test results confirmed differences in the level of physical activity and smoking [20].

In an additional study, the authors considered the need for a combination of anthropometric studies and functional tests in the analysis of the health of individuals involved in health-related activities [21].

Other researchers emphasize the importance of monitoring the morphofunctional performance of athletes to control the training process. The accuracy of the prediction can be improved through an integrated approach to assess the condition [22].

The use of dosed loads and functional tests is widely used in the practice of physical culture and sports. An analysis of the response of vegetative systems to loads is an important tool for monitoring the functional condition [23].

The design which was used (comparing the effect of health-related programs of selective and complex influence) is widely applied in scientific research. A similar design was used by Ponomareva [8]. The author compared changes in the level of physical activity and physiological parameters of students as a result of aerobics classes and traditional types of exercises. In the experimental group, a significant improvement in indicators characterizing physical development was determined: the level of physical qualities development, the condition of the cardiovascular system (pulse under load) and the respiratory system (inspiration time). The improvement trend was revealed in terms of heart rate recovery using a functional test. An increase in the speed of heart rhythm recovery confirms the tendency to expand the adaptive potential and functional capabilities of the cardiovascular system and the body in general.

A similar design was used in another study. The authors compared the health-related effects in healthy young women that could be noted after eight-week strength training and stretching exercises [24].

Ramirez-Campillo et al. used a similar design in the analysis of physical condition in women amateur football players. The authors compared the effect of similar training and a different focus [25].

The design used a comparison of a comprehensive health-improving program with narrowly focused exercises, confirmed the higher effectiveness of exercises in group 1. This is due to the fact that the combination of different types of activities allows you to provide a complex healing effect on the body.

The effectiveness of the health programs that were used is confirmed by the period of their implementation. This allows for achieving a long-lasting effect. Changes in the functional condition should be formed and fixed in 8 months of conducted classes.

Similar data are provided in a research in which the authors evaluated the effectiveness of aerobic exercise in women of a similar age group. After 4 months of training, no effects were found out. After 12 months, working efficiency improved, and the duration of recovery decreased [26].

It was confirmed that the eight-week Zumba training program had a positive effect on indicators (body composition, functional mobility assessment results, and dynamic balance parameters) in high school female students (15–17 years old) with a high body mass index [27].

The determined morphofunctional changes in the participants of group 1 reflect the complex effect of the workout classes. The combination in the training program of strength development, strength endurance, aerobic orientation, led to an increase in muscle strength, expansion of the adaptive potential of the cardiovascular and respiratory systems, and increased resistance to hypoxia. In this case, there is cross-adaptation, when integrated training leads to an increase in tolerance to physical activity of various kinds.

Similar results were obtained in tests which confirmed the positive influence of dance and strength fitness on the morphological parameters of women. The research showed the expansion of the adaptive-compensatory potential of the participants [28].

An increase in hand muscle strength indicators should be assessed as evidence of an increase in physical fitness. These indicators are used as indicators of health status, their decrease is assessed as the risk of developing disorders [29,30].

The nature of the changes in groups 1 and 2 is largely stipulated by the characteristics of the classes. Stretching had a pronounced health-related effect due to the optimization of physiometric indicators, body mass index normalization. However, this effect is not expressed in all participants as its severity is lower than in group 1. The conducted analysis allows for concluding that a comprehensive health-related program has a significantly higher functional effect than narrowly focused health classes.

Improving the goniometric indicators reflects the effect of fitness classes aimed at developing flexibility. The more pronounced changes in the participants of group 2 reflect the formation of specific adaptive changes in the musculoskeletal system. This is a reflection of the target effect of a narrowly specialized program.

Close results were obtained by authors of another research in which they define a pronounced selectivity of the effects of Pilates exercises on the body. The inclusion of such classes in training had a positive effect on ligament elasticity in young female volleyball players. No improvement was found in the explosive strength of the legs [31].

## 5. Conclusions

The dynamics of the morphofunctional indicators of the participants after the completion of workout classes testifies in favor of using the complex programs. Such a program has a more pronounced recreational effect. It intensifies the adaptive potential, the functional capabilities of the participants, and the optimization of the studied indicators. The complex program has a more multifaceted influence with the same time spent in fitness classes. This allows us to consider it as an adaptogenic and ergogenic factor.

Health-related classes have a positive effect on the morphological and functional condition of the second mature aged women. The comparative analysis of the dynamics of the studied criteria showed that a complex program has several advantages. The improvement of physiometric indicators and the optimization of the functional condition of the cardiovascular and respiratory systems was determined. This allows for predicting the increase of the adaptive potential of women. Improving training sessions of a specialized rehabilitation program (stretching) has a more specific and narrowly focused effect. This is confirmed by the increase in the amplitude of movements in the main joints.

## Figures and Tables

**Table 1 ijerph-17-08465-t001:** Morphofunctional and goniometric indicators of the experimental (1) and control (2) groups.

Indicators	1 Group		2 Group	
	The Beginning of the Experiment	The End of the Experiment	The Beginning of the Experiment	The End of the Experiment
Right Handgrip Strength Test (kg)	20.60 ± 0.74 ^†^	23.95 ± 0.76 ^‡^	18.04 ± 0.88	18.61 ± 0.91
Left Handgrip Strength Test (kg)	18.58 ± 0.87 ^†^	21.55 ± 0.73 ^‡^	13.96 ± 0.86	14.61 ± 0.86
Vital capacity (mL)	2692.50 ± 80.49	2880.00 ± 73.75 ^‡^	2634.78 ± 59.51	2652.17 ± 59.93
Body mass index, (kg/m^2^)	27.30 ± 0.88	25.18 ± 0.74	26.56 ± 1.10	26.33 ± 1.00
Systolic blood pressure (mm Hg)	121.38 ± 2.24	116.05 ± 1.65	124.43 ± 3.58	122.39 ± 3.31
Diastolic blood pressure (mm Hg)	76.63 ± 1.56	74.00 ± 1.24	80.22 ± 2.68	79.00 ± 2.54
Pulse pressure (mm Hg)	45.00 ± 1.43	42.05 ± 0.98	44.22 ± 1.55	43.39 ± 1.44
Resting heart rate (bpm)	77.83 ± 1.74 ^†^	71.23 ± 1.37 ^‡^	79.87 ± 2.32	79.35 ± 2.15
Stange test (s).	32.15 ± 2.18 ^†^	41.33 ± 2.61 ^‡^	27.35 ± 2.10	28.48 ± 2.07
Genci test (s).	18.88 ± 0.96 ^†^	24.85 ± 1.06 ^‡^	19.00 ± 1.35	19.87 ± 1.30
Right Shoulder flexion (degrees)	163.40 ± 2.03 ^†^	173.80 ± 1.38	159.52 ± 2.62 ^†^	171.04 ± 2.18
Left Shoulder flexion (degrees),	162.08 ± 1.91 ^†^	173.30 ± 1.28	159.52 ± 2.35 ^†^	171.39 ± 1.98
Right Shoulder Extension (degrees)	37.10 ± 1.26 ^†^	44.75 ± 1.30	34.26 ± 1.91 ^†^	42.57 ± 1.97
Left Shoulder Extension (degrees)	37.08 ± 1.39 ^†^	44.50 ± 1.32	34.39 ± 1.93 ^†^	42.39 ± 1.94
Right Shoulder abduction (degrees)	158.65 ± 2.37 ^†^	169.00 ± 1.86 ^‡^	168.78 ± 1.58 ^†^	178.30 ± 0.90
Left Shoulder abduction (degrees)	157.70 ± 2.29 ^†^	168.48 ± 1.78 ^‡^	169.13 ± 1.55 ^†^	178.35 ± 0.93
Right Hip flexion (Straight Leg) (degrees)	79.93 ± 1.60 ^†^	88.05 ± 1.27 ^‡^	80.26 ± 2.06 ^†^	93.13 ± 2.15
Left Hip flexion (Straight Leg) (degrees)	79.45 ± 1.57 ^†^	87.58 ± 1.22 ^‡^	80.57 ± 1.93 ^†^	93.48 ± 1.95
Right Hip extension (degrees)	14.20 ± 0.78 ^†^	19.20 ± 0.78	16.17 ± 1.21 ^†^	22.35 ± 1.48
Left Hip extension (degrees)	13.98 ± 0.74 ^†^	19.10 ± 0.76 ^‡^	16.26 ± 1.15 ^†^	22.35 ± 1.40
Torso Flexion (degrees)	120.13 ± 2.44 ^†^	130.40 ± 2.21 ^‡^	126.96 ± 2.81 ^†^	138.30 ± 2.09

Note: ^†^ differences between the beginning of the experiment and the end of the experiment are significant (*p* < 0.05); ^‡^ differences between groups are significant (*p* < 0.05).

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
