# Peer review of "Influence of Health Related Fitness on the Morphofunctional Condition of Second Mature Aged Women"

_ijerph, 2020, doi:10.3390/ijerph17228465_

Round 1
Reviewer 1 Report
Authors must make structural and formal improvements to the manuscript. A new review of the articles used in the introduction and discussion is recommended.
- Improve introduction and discussion. The wording of this section does not follow an adequate methodology. The same structure is repeated in most of the paragraphs (Page 2, line 47: Nazari and MacDermid emphasize…).
- Material and methodology: the sections must be numbered (2.1, 2.2, etc.).
- In table 1, the indicators could be described as right and left and the unit in parentheses. Example: left manual dynamometry (kg).
- Tables must be self-explanatory. The notes in the lower margin of the table do not adequately explain the relationship between the results obtained. What is dynamics?.
- There must be a space between the end of the line and the quote.
- The type of strength and flexibility training performed by the sample subjects should be described.
- In the introduction and discussion, some important studies on the assessment of physical condition in youth and adults have not been included. Some examples are:
- Wellbeing as a Protective Factor of Adolescent Health. The Up & Down Study.
- Mediation role of cardiorespiratory fitness on the association between fatness and cardiometabolic risk in European adolescents: The HELENA study.
- Association of Patterns of Moderate-to-Vigorous Physical Activity Bouts With Pain, Physical Fatigue, and Disease Severity in Women With Fibromyalgia: the al-Ándalus Project.
- Muscle strength field-based tests to identify European adolescents at risk of metabolic syndrome: The HELENA study.
- Exercise Training as a Treatment for Cardiometabolic Risk in Sedentary Adults: Are Physical Activity Guidelines the Best Way to Improve Cardiometabolic Health? The FIT-AGEING Randomized Controlled Trial.
- Cardiorespiratory fitness, muscular strength, and obesity in adolescence and later chronic disability due to cardiovascular disease: a cohort study of 1 million men.
- Grip strength cutpoints for youth based on a clinically relevant bone health outcome.
Author Response
We appreciate dear reviewers for the comments. Below we have written changes that were in the article. All our changed we have singled out by yellow colour
-
We have made the corresponding changes in sections Introduction and Discussion.We have deleted text and reference on a source in References (Page 2, line 47: Nazari and MacDermid emphasize…).
- We have made the corresponding changes in section Material and Methods
- We have made the corresponding changes in Table 1.
- We have made the corresponding changes in section Discussion
- We have made the corresponding changes in section References
Reviewer 2 Report
In their study, the authors tried to compare the influence of different programs related to health-related fitness on the morphofunctional conditions of women in the second stage of their life. They concluded that improving training sessions of a specialized rehabilitation program, such as stretching, leads to a more specific focused effect, mainly in the amplitude of movements in the main joints.
The study and its results are logical, but in my opinion, the authors should have better described a series of conditions and analyzed other physical activities. For instance, their study has been carried out in the second age mature women, but they did not indicate what number of these women had work and the kind of it; neither they indicated the number of menopausic women in each group or they had children or not. Moreover, they compared the physical activity carried out by three different methods versus only stretching. It would be useful to analyze aerobics (2 sessions/week) plus stretching (1 session/week), and strength fitness(2 sessions/week) plus stretching (1 session/week) versus stretching (3 sessions/week) in order to establish what is the best exercise to improve health, besides the study they made.
The Introduction describes previous studies carried out by other groups. Apart of avoiding to type the name of every author of these studies (only the reference would be needed), they affirm that moderate-intensity endurance training causes a favorable adaptation of skeletal muscles due to an increase in mitochondria; an explanation of it (number of mitochondria? increase in aerobic or anaerobic metabolism...) and a reference is needed here.
In the section Methods (amplitudes, degrees) the authors repeat right and left for each amplitude analyzed. They could write bilateral shoulder flexion, hip flexion, etc... This would reduce the size of the manuscript without affecting its meaning.
In the section Discussion (line 163), there is a phrase without any sense: The various indicators and tests used for this purpose. What want the authors say?.
Redaction is poor and the English needs to be revised, as well as some typographic errors.
Author Response
We appreciate dear reviewers for the comments. Below we have written changes that were in the article. All our changed we have singled out by yellow colour.
- Respectively, we have added in Reference sources 1-5.
- We agree with the following that it can be used a description of shoulder position as a reviewer proposed. But, in biomechanics is also used a pointed by us position of shoulder.
- We agree with Comment 4"In the section Discussion (line 163), there is a phrase without any sense: The various indicators and tests used for this purpose. What want the authors say?"
. and delete this phrase. We have made the corresponding changes in section Discussion
Round 2
Reviewer 1 Report
- Change font: "2.1 Participants" by 1. Participants (line 84).
- Include a line space in each statement:
"2.1. Participants"
"The study involved…"
- Change (line 131): “Note – 1- differences…” for Note: 1 differences…
Author Response
Thank you for your review. The article has been correctedReviewer 2 Report
The authors have made changes in their manuscript according to the suggestions made by this reviewer. However, the style of the manuscript is imperfect and should be corrected. For instance, they should avoid writing so many references to other studies beginning with the names of the authors of these manuscripts, mainly in the Introduction. For example Osipov et al....., Kokun et al...., etc. It would be easier to write the description of the study followed by the reference [6], [9]. This is repeated in most of the text.
The English need a strong correction.
Author Response
Thanks for your review. The article has been corrected. Additionally, the English language was improved with the help of the Native Speaker